# Dietary Fatty Acids and Host–Microbial Crosstalk in Neonatal Enteric Infection

**DOI:** 10.3390/nu11092064

**Published:** 2019-09-03

**Authors:** Candice Quin, Deanna L. Gibson

**Affiliations:** 1Department of Biology, Okanagan Campus, University of British Columbia, Okanagan Campus ASC 386, 3187 University Way, Kelowna, BC V1V 1V7, Canada; 2Department of Medicine, Faculty of Medicine, University of British Columbia, Kelowna, BC V1V 1V7, Canada

**Keywords:** dietary fatty acids, saturated fatty acids, monounsaturated fatty acids, polyunsaturated fatty acids, formula, enteric infection, infant nutrition

## Abstract

Human milk is the best nutritional choice for infants. However, in instances where breastfeeding is not possible, infant formulas are used as alternatives. While formula manufacturers attempt to mimic the performance of human breast milk, formula-fed babies consistently have higher incidences of infection from diarrheal diseases than those breastfed. Differences in disease susceptibility, progression and severity can be attributed, in part, to nutritional fatty acid differences between breast milk and formula. Despite advances in our understanding of breast milk properties, formulas still present major differences in their fatty acid composition when compared to human breast milk. In this review, we highlight the role of distinct types of dietary fatty acids in modulating host inflammation, both directly and through the microbiome-immune nexus. We present evidence that dietary fatty acids influence enteric disease susceptibility and therefore, altering the fatty acid composition in formula may be a potential strategy to improve infectious outcomes in formula-fed infants.

## 1. Introduction

The neonatal period represents a critical time in mammalian life, particularly with respect to nutritional programming of the gut microbiome and immunity required to respond to infectious diseases yet stave off chronic inflammatory conditions like allergies and autoimmune disease. The relationship between infant health and breastfeeding has been recorded for thousands of years and across numerous civilizations [1]. Dating back to the turn of the 19th century, observations that breastfed infants had significantly lower rates of mortality and infectious diarrhea compared to “bottle-fed” infants, spiked research aimed at deciphering the biologically active components in breast-milk. In the 1950s, improved hygiene and nutritional content of formula resulted in better weight gain and a dramatic reduction in infant mortality [2]. As a result, many believed that formula feeding was analogous to breastfeeding, a notion that was reinforced by aggressive marketing of formula companies. This resulted in a steady decline in breastfeeding in the 1970s [3] with exclusive breastfeeding rates going from 90% in the 20th century to approximately 42% in the 21st century, globally [4,5]. However, despite the significant improvements to the quality of infant formula, studies still show that formula-fed infants are more susceptible to enteric infections [6,7], a statistic likely attributed to compositional differences between the two nutritional sources. A difficult challenge in mimicking the composition of breast milk stems from the complexities of breast milk itself. Breast milk composition differs at various times of the day, throughout feeding, and at distinct stages of lactation. Moreover, while human milk protein and carbohydrate concentrations appear to be relatively static in breast milk, fatty acid contents vary considerably and may be influenced by maternal dietary intake [8]. Despite these challenges, replicating breast milk lipid composition in formula has the potential to minimize the differences in disease risk between formula and breastfed infants. Triacylglyerols contribute 98% to 99% of human milk fat. The properties of triglycerides are influenced by fatty acid composition. Formula companies generally derive their fatty acids from vegetable sources, which differ greatly from the fatty acids found in human breast milk [9]. Different structures of fatty acids alter the host inflammatory process [10], including defensive inflammatory responses following infection [11]. This affects the severity of disease, making fatty acids an important consideration during the neonatal period. The mechanism by which various fatty acids alter enteric disease susceptibility may be through direct fat–host interactions or through chemical dialogue with the intestinal microbiome. This review will consider the relationship between the microbiome and immune development in infants and discuss the involvement of dietary fatty acids in modulating intestinal microbial communities. The various mechanisms by which the three main structural groups of fatty acids including saturated, monounsaturated and polyunsaturated fatty acids alter host inflammatory processes will be considered here. Finally, the evidence that dietary fatty acids and host-microbial crosstalk alter outcomes of prevalent enteric diseases in infants will be reviewed. Due to the paucity of literature on these interactions in human neonates, many of the studies discussed here are based on animal models or adult human research. Overall, this review serves to develop strategies to improve infectious outcomes in formula-fed infants by understanding the fatty acid-microbe-immune dynamic.

## 2. Methods 

### 2.1. Literature Search Strategy

For this literature review, we searched PubMed (1966 to 2019), Web of Science (1864–2019), Google Scholar and hand searches for primary studies investigating the effects of dietary fatty acids on the microbiome, immune function and common enteric infections in infants. The search strategy was based on a clear and careful selection of key words including combinations of: diet* AND (fatty acid* OR fat* OR polyunsaturated* OR monounsaturated* OR saturated* OR fish oil*) AND (immune* OR inflammation* OR innate* OR adaptive*) AND (enteric disease* OR rotavirus* OR Escherichia coli* OR Shigella* OR Salmonella* OR Vibrio cholerae* OR Campylobacter jejuni* OR Citrobacter rodentium*) AND (bacteria* OR commensal* OR microbiome* OR bacteriome* OR microorganisms* OR flora*).

### 2.2. Inclusion and Exclusion Criteria

The included studies had to report the effects of a specified fatty acid diet on either the microbiome, immune responses or enteric disease outcomes. Interactions between dietary fatty acids and enteric diseases were restricted to animal studies reporting on morbidity and mortality outcomes. Studies were excluded if the type of dietary fat consumed was not specified. Microbiome studies were also excluded if they did not report on the gastrointestinal microbiome.

## 3. Microbial Programming of the Infant Immune System

Humans have evolved a symbiotic relationship with their intestinal microbiome, a vast and complex microbial ecosystem composed primarily of bacteria, but also contains fungi, viruses and archaea. The microbiome lies at the interface of the intestinal and external environment, forming a bridge between intestinal epithelial cells and dietary antigens. Here, the commensal bacteria play a crucial role in nutrient absorption, accounting for up to 10% of the dietary energy in mammals [12]. However, in the past decade researchers have come to appreciate that this mutually beneficial relationship extends beyond food digestion. An ever-increasing body of evidence indicates that exposures to commensal microorganisms early in life are essential for the proper development and function of the mucosal immune system. While the interactions between commensal bacteria, intestinal morphology and immune development have been well-reviewed [13,14,15], here we provide an overview to highlight how dietary fatty acids synergize with this process. 

## 4. Microbiome and Immune Development

The human intestine is the largest immunologic organ in the body and serves as the first line of defense against enteric pathogens. At parturition, the intestinal epithelium in newborn infants is immature and allows ingested foreign particles and bacteria to readily gain access to the bloodstream and lymphatic system [16]. While there is considerable debate surrounding a so-called “leaky gut” in infants, increased permeability actually facilitates communication between the commensal microbiome and the host, but detailed mechanistic insights explaining these interactions remain unclear. Nonetheless, studies show that signals from commensal bacteria are transmitted to intestinal epithelial cells and trigger maturation of the mucosal immune system [17]. Intestinal epithelial cells act as a physiological barrier against invading pathogens. During infection, intestinal epithelial cells recognize microbe-associated molecular patterns through interactions with host Toll-like receptors. Following recognition, intestinal epithelial cells secrete inflammatory cytokines, which promote inflammation and attract leukocytes to the site of infection where they act to neutralize the pathogen. This cascade of inflammatory events is impaired in germ-free (sterile) animals. Compared to animals reared under specific pathogen-free conditions, germ-free animals have fewer Peyers Patches, smaller lymphoid follicles [18], altered epithelial morphology [19] and abnormal germinal centers in the small intestine [20]. Germ-free animals also exhibit decreased intestinal T cell subsets and reduced numbers of lymphocytes expressing αβ T cell receptors [21]. Reconstitution with commensal gut bacteria can restore many of these immunological abnormalities [21,22,23]. However, emerging evidence suggests that some of these corrections are age-dependent and that there is a critical “window of opportunity” during infancy where colonization must occur to ensure proper intestinal immune development [24]. Effective immune maturation also depends on colonization with host-adaptive microbiota [25] and current evidence suggests that different types of bacteria elicit different responses. For example, while the Gram-negative *Bacteroides thetaiotaomicron* induces expression of RegIIIγ, an antimicrobial peptide, the Gram-positive *Bifidobacterium longum* does not [26,27]. Thus, immune development relies on the presence of specific taxa within the commensal microbiota and their overall community structure. Knowledge of factors that modulate community structure in formula-fed infants will, therefore, be important in positively shaping similar immune development to breastfed infants. 

## 5. Microbiota and Dietary Fatty Acids

The intestinal microbiota resides at the interface of the external environment and the host, allowing for direct interactions with dietary antigens. Of all the exogenous factors implicated in contributing to the phylogenetic makeup of the microbial community, dietary exposures are key [28]. During the first day of life, breastfed and formula-fed infants have comparable microbial communities containing members of the family Enterobacteriaceae, as well as *Streptococcus, Staphylococcus* and *Enterococcus* spp. [29]. However, by the end of the first week formula-fed infants have significantly less *Bifidobacteria* spp. than breastfed infants [30,31,32,33] and increased species richness and diversity [34] resembling that of an adult [35]. As commensal bacteria have profound effects on immune development, deviations in intestinal colonization from the breastfed ‘norm’ may predispose formula-fed infants to have higher incidences of infections from enteric pathogens [36,37]. Despite the many publications about the effects of dietary carbohydrates, the impact of dietary fatty acids on intestinal colonization is less well defined. There are several reports that “high-fat” diets disrupt the gut microbiota by promoting a decrease in *Bacteroidetes* and a corresponding increase in *Firmicutes* and *Proteobacteria* [38,39]. However, dietary fatty acid type, rather than the total number of calories, plays a role in susceptibility to enteric infectious disease [11], making it difficult to interpret many high-fat studies without considering the type of fat and controlled comparisons. Furthermore, in many rodent microbiome studies, fatty acids are used at concentrations which may not be physiologically relevant or do not represent human intake [40]. Nevertheless, there is enough evidence showing that dietary fats can markedly alter gut microbial communities.

There are three main structural categories of fatty acids. These include saturated fatty acids, monounsaturated fatty acids and polyunsaturated fatty acids, so named by the number of double bonds in the acyl chain. The position of the double bond in polyunsaturated fats further distinguishes between omega-3 polyunsaturated fatty acids and omega-6 polyunsaturated fatty acids. A series of recent studies investigating the impacts of dietary fat type on the gastrointestinal microbiome in C57BL/6 mice have shown that dietary fatty acid type distinctly affects gut microbial diversity [41]. Diets rich in omega-6 polyunsaturated fatty acids were associated with inflammatory-inducing microbial blooms of Enterobacteriaceae, Segmented Filamentous Bacteria and *Clostridia* spp. [42]. These bacterial communities were predicted to have enhanced virulence and pathogenicity potential and associated with host proteins involved in inflammation, oxidative stress and barrier dysfunction [41]. Interestingly, supplementing omega-6 polyunsaturated fatty acid diets with omega-3 polyunsaturated fatty acids enriched anti-inflammatory microbes such as *Lactobacillus* and *Bifidobacteria* spp. [42]. Like omega-6 polyunsaturated fatty acids, consumption of saturated fats resulted in a host–microbe relationship indicative of inflammation, and shared similarities between dominant bacterial species. This observation is consistent with results observed in genetically susceptible IL-10 knockout mice where milk-derived saturated fat promoted the expansion of inflammatory-inducing *Bilophila wadsworthia* and associated with increased pro-inflammatory responses [43]. Similarly in piglets, the addition of both milk fat and milk fat globule membrane fragments to infant formula increased abundances of *Escherichia/Shigella* and *Klebsiella* genus and altered the dynamic of the immune system maturation to better resemble breastfed piglets compared to vegetable-derived fat [44]. However, while both saturated fats and omega-6 polyunsaturated fats enable inflammatory-inducing bacterial populations to proliferate, rare bacterial species were unique to the milk-fat diet and associated with different functional characteristics compared to omega-6 polyunsaturated fats. For instance in C57BL/6 mice, while saturated fat promoted intestinal inflammation, there was also a compensatory protective response in both the host and microbiome through increased host sirtuin signaling pathways and microbial production of beneficial short-chain fatty acids [41]. In contrast, monounsaturated fatty acids promoted a microbial community, which clustered separately from saturated and omega-6 polyunsaturated fatty acids and was more similar to a low fat (9% energy from corn oil) diet. Collectively, this data shows that dietary fatty acids change the intestinal ecosystem by enabling certain bacterial populations to proliferate and dampening the expansion of others. 

Fatty acids can normalize the microbiome and other aspects of intestinal immune development in formula-fed infants. Milk fat globule membrane, a unique fat component of breast milk, fed to rat pups as a fortified formula affected intestinal crypt depth, epithelial cell subsets, epithelial cell proliferation and intestinal microbial communities in a similar manner to pups reared on mother’s milk [45]. In contrast, the control formula group had significant deficits in intestinal development. When challenged with *Clostridium difficile* toxins, the fortified formula afforded significant protection from mucosal damage when compared to pups fed the control formula. This study thus directly links the gut microbiome, dietary fatty acids and enteric disease susceptibility and suggests a strategy to improve infectious outcomes in formula-fed infants. Indeed, a double-blind randomized controlled trial found bovine milk fat globule membrane protein lowered incidences of diarrhea in infants [46]. Together these studies show that dietary fats can markedly alter conditions for gut microbial assemblage. 

## 6. Fatty Acids and Inflammation

Although it has been known for decades that dietary fatty acids can directly alter mammalian inflammatory responses [47], recent studies have only just begun to appreciate that the saturation index between fatty acids has profound effects on inflammation (Figure 1). As described above, immune cells secrete a wide variety of chemical signals during inflammation. Many of these inflammatory mediators such as prostaglandins, leukotrienes and thromboxanes are derived from the omega-6 polyunsaturated fatty acid, arachidonic acid. While there is strong evidence supporting the link between arachidonic acid-derived eicosanoids and inflammation, it is important to note that not all arachidonic acid-derived eicosanoids are pro-inflammatory. For instance, lipoxin A_4_ plays an important role in inflammation resolution following infection by clearing neutrophils from the site of infection and by inhibiting NFκB activation [48,49]. Nevertheless, arachidonic acid-derived eicosanoids are central in inflammatory responses.

In stark contrast to the pro-inflammatory lipid mediators formed from omega-6 polyunsaturated fatty acids, omega-3 polyunsaturated fatty acids have long been considered “anti-inflammatory” [10]. Increased consumption of omega-3 polyunsaturated fatty acids substitutes arachidonic acid in immune cell membranes. While omega-3 polyunsaturated fatty acids such as docosahexaenoic acid and eicosapentaenoic acid are still viable substrates, the eicosanoids they produce are less biologically active than those derived from arachidonic acid, resulting in a comparably lower inflammatory state. In addition to this action, several other mechanistic insights have been identified which suggest omega-3 polyunsaturated fatty acids have anti-inflammatory properties. For example, omega-3 polyunsaturated fatty acids have been shown to reduce inflammatory cytokine production following endotoxin stimulation [50,51] and can limit the activation of NFκB [52]. Omega-3 polyunsaturated fatty acids can also inhibit migration of immune cells towards chemoattractants such as leukotriene B4 and bacterial peptides [53,54,55] and lower the expression of intracellular adhesion molecule 1 on stimulated blood monocytes, ex vivo [56]. Ultimately, this prevents the infiltration of leukocytes to the site of inflammation. Additionally, omega-3 polyunsaturated fatty acids can promote resolution of inflammation through specialized pro-resolving mediators such as resolvins and protectins (Reviewed [57]). While these specialized pro-resolving lipid mediators resolve inflammation and protect against chronic inflammatory conditions such as colitis [58], neuroinflammation [59] and arthritis [60] there is a paucity of research on how they impact enteric disease risk. Therefore, this review will focus on the impacts of dietary fatty acids instead of their downstream mediators. 

In contrast to the effects of polyunsaturated fatty acids on inflammation, the mechanism by which saturated or monounsaturated fatty acids influence host inflammation remains unclear. Limited studies suggest that saturated fatty acids increase host inflammation. For example, laurate and palmitate have both been shown to activate Toll-like receptors 2 and 4. These immune cell surface receptors activate innate immune signaling pathways which are critical for host defense against pathogens. Collectively, these findings highlight that the impact of fatty acid in the context of inflammation is complex and that it is important to consider fat type when examining host responses to enteric pathogens.

## 7. Dietary Fatty Acids, the Microbiome and Enteric Disease in Infants

Diarrheal diseases are the second leading cause of death in children under the age of five, accounting for 525,000 children deaths every year [61,62]. Diarrhea is a symptom of an infection in the gastrointestinal tract, which can be caused by a variety of bacterial, viral and protozoan organisms. The most prevalent etiological agents include rotavirus, *Escherichia coli* strains, *Shigella* spp. *Salmonella enterica* subspecies, *Vibrio cholerae,* and *Campylobacter jejuni* [63]. Diarrheal deaths are preventable and often treatable, yet only 44% of children with diarrhea receive the recommended treatment due to the lack of clean water and oral rehydration therapy [64]. Therefore, determining effective interventions to prevent and control diarrheal diseases in infants remains a critical undertaking. Infectious disease susceptibility during infancy is intricately linked to malnutrition [63,65,66]. Nutritional deficiencies during pregnancy can be partially compensated for by breastfeeding. However, if breastfeeding is not feasible, inadequate or unsuitable, formula becomes imperative to avoid nutritional deficiency in children born to poorly-nourished mothers. While remarkable improvements have been made to the safety of infant formula over the past few decades, epidemiologic studies continue to show that formula-fed infants are more susceptible to diarrheal diseases [67]. One plausible explanation for the differences in disease risk is the different fatty acid profiles between the two nutritional sources [68]. Given that different types of dietary fat illicit varying host inflammatory responses and can impact intestinal microbial communities, it stands to reason that consumption of various fats also impact enteric disease susceptibility. 

Ethical considerations preclude infecting infant populations to examine the relationship between dietary fatty acids and disease susceptibility. As such, our understanding of these interactions is often restricted to animal studies, which have limitations. For example, enterohemorrhagic *Escherichia coli* (EHEC) and enteropathogenic *E. coli* (EPEC) are global causes of diarrhea and mortality in children. The formation of attaching and effacing lesions distinguishes these bacteria from other commensal *E. coli* strains commonly found in the infant’s gut. However, these pathogens do not readily infect most laboratory animal species, including mice. As a result, the related pathogen *Citrobacter rodentium* which readily colonizes mice and shares virulence factors with human EPEC is used as a surrogate approach [69]. Understanding the pathogenesis of *C. jejuni,* and *Salmonella enterica* serovar Typhimurium infections are similarly limited by a lack of relevant animal models. *C. jejuni* does not naturally cause disease in mice and therefore genetically manipulated animals such as IL-10 deficient mice are used [70]. *Salmonella* Typhimurium likewise has little natural colonization in the intestine in mice but does cause system typhoid like disease. Administration of streptomycin prior to *Salmonella* infection can induce enteritis [71]; however, this negates any interactions with the intestinal microbiome. While there are novel models being developed to study salmonellosis in the absence of anti-biotic pre-treatments [72], currently no studies have used this new approach to study the relationship between dietary fatty acids, the microbiome and enteritis susceptibility. Thus, we report only on systemic *Salmonella* infections administered in the absence of streptomycin. While acknowledging the limitations of animal studies, here we review evidence that different types of dietary fatty acids impact disease risk to the most common pathogens causing diarrhea in infants. 

## 8. Saturated Fatty Acids

Saturated fatty acids comprise the major portion of breast milk, making up 34% to 47% of the total fat component [9]. Formulas containing saturated fats from dairy were widely used at the beginning of the 20th century, but their global use has diminished [9]. Cow-milk based formulas today use cow’s milk proteins, but vegetable fat sources to meet the minimum content of linoleic and alpha-linoleic acid [73]. For decades the consumption of saturated fat was thought to increase cardiovascular risk and one of the proposed means to reduce cardiovascular disease in adults was to reduce consumption of saturated fat in infants. Throughout the 1960s and early 1970s a number of physicians recommended the use of skim milk [3,74] and formulas prepared with isolated soy protein became commercially available as a nonmilk-based alternative. However, there is little evidence that restricting fat in infants is beneficial and the American Academy of Pediatrics recommended against these dietary changes believing the benefits of dairy fat outweighed the risks [75]. Early observational studies supported the benefits of saturated fats and found that children consuming whole dairy milk were five times less likely to develop acute gastrointestinal illness than children consuming a low-fat alternative [76]. Today, studies continue to question whether dietary saturated fats should be restricted. Recent meta-analyses of observational studies have shown no association between saturated fat and cardiovascular disease risk [77,78]. There is some evidence that diets rich in saturated fatty acids from palm oil reduce the abundance of bacteria in the Bacteroidetes phylum such as *Bacteroides* spp., which increases the Firmicutes to Bacteroidetes ratio [79,80]. An increased Firmicutes to Bacteroidetes ratio has been theorized to predispose infants to future obesity [81] supporting the cardiovascular risk theory. Counter to this, Walters et al. show in a meta-analysis that this ratio is not significantly different between obese and normal-weight adults [82] and conclude that the Firmicutes to Bacteroidetes ratio may not be the most robust marker for obesity [83]. Moreover, saturated fatty acid from animal fat has been shown to increase *Bacteroides* spp. in human stool [84] suggesting that the source of saturated fat is an important consideration. Irrespective of whether saturated fat is a risk factor for cardiovascular health and obesity, evidence suggests saturated fatty acids have either a positive or neutral effect on enteric disease outcomes.

Rotavirus is the leading cause of life-threatening diarrhea in babies. In 2016, the World Health Organization estimated a global 215,000 deaths in children under the age of five resulting from rotavirus infection [85]. Early studies on the interactions between fatty acids and viruses showed that human milk fats have antiviral effects [86] whereby medium-chain saturated fatty acids were highly active against herpes simplex viruses. To see if dietary fatty acids had antiviral effects against rotavirus, 3-week-old BABL/c mice were fed isocaloric diets rich in saturated fat from butter, monounsaturated fat from olive oil or omega-6 polyunsaturated fat from corn oil, for one week. The seven-day-old neonatal mice were then orally challenged with epidemic diarrhea of infant mice (EDIM) rotavirus. At 48 h post-inoculation, 72.7% of those on corn oil, 46% of those on olive oil and 50% of the controls but none of the butter group, developed diarrhea suggesting a protective effect from saturated fat and that unsaturated prompted viral infection [87]. 

In addition to studying the effects of fatty acids on rotavirus, several bacterial pathogens have been examined. EPEC is a major cause of persistent diarrhea in infants. Progress is being made in unraveling the relationship between dietary fatty acids and EPEC susceptibility using the *C. rodentium* model. One post-natal study examining the effects of anhydrous milk, rich in saturated fat, olive oil and corn oil on the pathogenesis of *C. rodentium* in C57BL/6 mice found no differences in mortality between the three dietary groups [11]. There was, however, increased mucosal damage in the milk fat and corn oil groups when compared to the olive oil-fed mice. This milk-fat associated intestinal mucosal damage has been observed in clinical studies of formula-fed infants consuming dairy milk [88]. Interestingly, the addition of fish oil to these background fat diets resulted in significant mortality in mice fed corn oil but not milk fat. This was attributed to an increase in compensatory protective responses in the milk fat group such as upregulated expression of intestinal alkaline phosphatase. Intestinal alkaline phosphatase is a mucosal defense factor that plays a critical role in intestinal health through interactions with the commensal microbiota, diet and the host epithelium [89]. Mice fed milk fat diets with fish oil also had the largest induction of protective short-chain fatty acids in the gut when compared to diets rich in olive oil or corn oil [11]. This is also likely due to interactions between dietary fats and the microbiome. Traditionally, nutritional research has focused on single nutrients; however, this research recognizes that total diet must be considered because there are interdependent relationships among dietary components. 

Similar to EPEC, some strains of *V. cholerae* causes severe diarrhea in infants. While a vaccine to prevent infection with cholera exists, it is not always effective and other preventative strategies are needed. One study evaluating the antibacterial properties of medium-chain monoacylglycerides and free fatty acids found that concurrent administration of monocaprin, a monoglyceride of the saturated fatty acid capric acid, with *V. cholerae* can significantly reduce viable *V. cholerae* in the gastrointestinal tract of mice. However, while administration of monoacyglycerides appears to reduce colonization of *V. cholerae,* it has little effect once the pathogen is established [90]. Unfortunately, the mice in this study were treated with streptomycin to predispose them to bacterial colonization. Because these mice were treated with streptomycin, any synergistic role of the microbiome would be eliminated. Therefore, new models are required to understand crosstalk between dietary fatty acids, the host and the microbiome during *V. cholerae* infections. 

*Salmonella* disease causes 93.8 million illnesses and 155,000 deaths annually [91]. Despite the fact that licensed *Salmonella* vaccines are available, these vaccines are not used in countries that most need them [92] and other strategies to reduce *Salmonella* risk are being explored. One proposed mechanism to reduce *Salmonella* mortality is through dietary fat modifications. In support of this, Chang et al. found that feeding female Swiss Webster mice hydrogenated coconut oil (rich in saturated fatty acids) for one week decreased mortality from systemic infection when orally challenged with *Salmonella* Typhimurium compared to mice fed a corn oil (rich in omega-6 polyunsaturated fatty acids) or fish oil (rich in omega-3 polyunsaturated fatty acids) diet [93]. However, a contradicting study published around the same time concluded that fat from differing sources had no effect on outcomes of systemic infection [94]. In this study, female CF1 mice administered *Salmonella* Typhimurium intraperitoneally showed no differences in survival among dietary groups fed coconut oil, oleic acid (rich in monounsaturated fat), safflower oil (rich in omega-6 polyunsaturated fatty acids) or fish oil (rich in omega-3 polyunsaturated fatty acids). Discrepancies between these two findings could be attributed to differences in mouse genetics or sites of infection and present an opportunity for future research. 

The protective attributes of saturated fat-based diets may be increased by the addition of live bacterial cultures. One study investigating the role of saturated fat-based diets in *Salmonella* Typhimurium infections found that yogurt-based diets are more protective than milk-fat based diets. In this study, mice were placed on a yogurt or milk-based diet for four weeks and challenged intragastrically with *Salmonella* in the absence of antibiotics [95]. The authors found that yogurt improved both local and systemic immunity and attribute this enhanced protection to the bacteria in the yogurt. This idea was supported by a separate study which showed that mice harboring low complexity gut microbiota are unable to clear *Salmonella* Typhimurium following administration with streptomycin [96]. By transferring a normal complex microbiota to these mice, colonization resistance and clearance are restored. These findings underscore the importance of dietary fatty acids and host-microbial crosstalk and emphasize the need for models that do not require pre-treatment with antibiotics. 

Collectively, these studies demonstrate that saturated fat-based diets promote robust inflammatory responses to enteric disease, contributing to survival. While this process initially causes inflammatory induced damage to the epithelium, it is followed by compensatory protective responses. These protective responses are intricately linked with the host microbiome and highlight the importance of understanding fatty acids and host-microbial interactions. 

## 9. Monounsaturated Fatty Acids

Monounsaturated fatty acids are the second more prevalent fatty acid component in breast milk, ranging from 31% to 43% of the total fat [9]. Like saturated fats, monounsaturated fatty acids appear to be protective against diarrheal diseases. As mentioned above, high-fat diets rich in olive oil are protective against *C. rodentium* [11]. Not only do these mice survive the infection but also, they show limited infection-induced pathology. One mechanism by which monounsaturated fats are proposed to reduce pathology is by directly limiting pathogen virulence traits. Enteric pathogens have evolved a remarkable array of virulence traits which enables them to colonize the intestinal epithelium and escape host defenses. Having the ability to reduce virulence traits through dietary interventions presents an appealing strategy for minimizing enteric disease risk. Controlled cell-culture experiments have shown that oleic acid, the monounsaturated fatty acid in olive oil, downregulates gene expression of *espA* and *tir*, two early virulence factors involved in EPEC attachment to the intestinal epithelium. Attachment assays confirmed that 40% fewer EPEC were attached to Caco-2 cells in the presence of oleic acid, compared with linoleic acid. Thus, it was concluded that diets rich in olive oil protect against enteric infection, in part, by decreasing pathogen attachment to the intestinal epithelium. Mice fed olive oil also have increased expression of IAP in the epithelium compared to mice fed corn oil or milk fat [11]. A separate study shows similar protective responses of olive oil against lipopolysaccharide (LPS)-induced endotoxic shock [97] but contributed the protection to a reduced magnitude of inflammatory processes. C57BL/6J mice were fed diets rich in olive oil, canola oil, sesame oil or soybean oil for 6 weeks and then injected intraperitoneally with bacterial LPS to induce endotoxic shock. Mice fed olive oil had reduced neutrophil accumulation and lower levels of inflammatory mediators compared to mice fed the other diets. Interestingly, of the four diets tested, olive oil-fed mice were the most resistant to endotoxic shock and had the highest rate of survival. While this study suggests olive oil may be protective against bacterial pathogens, intraperitoneal injections do not allow for interactions with the intestinal microbiota. Overall, more research is needed to clarify the mechanisms by which monounsaturated fats protect against diarrheal diseases. 

## 10. Polyunsaturated Fatty Acids

Despite making up the lowest proportion of breast milk (12% to 26% omega-6 polyunsaturated fatty acid and 0.8% to 3.6% omega-3 polyunsaturated fatty acid) [9], polyunsaturated fatty acids have been the most extensively studied. Two polyunsaturated fatty acids, linoleic acid and alpha-linolenic acid are considered essential because they cannot be synthesized by the body. These fatty acids are respective dietary precursors for arachidonic acid and eicosapentaenoic acid, which is further converted to docosahexaenoic acid. However, the synthesis pathways required for these conversions involve several elongation and desaturation chemical reactions, which is less efficient than dietary uptake. For this reason, consumption of food containing non-essential fatty acids is recommended and formulas with arachidonic acid and docosahexaenoic acid are now commercially available. Current guidelines for the levels of polyunsaturated fats in infant formula aim to avoid a high linoleic (omega-6 polyunsaturated fatty acid) to alpha-linolenic (omega-3 polyunsaturated fatty acid) ratio [9]. This is because extremely high levels of omega-6 polyunsaturated fatty acids in formula may have untoward effects due to their pro-inflammatory functions discussed above. The effects of omega-3 polyunsaturated fatty acids in rodent studies appear to depend on the ‘background’ dietary fat. When combined with omega-6 polyunsaturated fatty acids, omega-3 polyunsaturated fatty acids encumber the host’s ability to combat enteric infection resulting in increased mortality. This is not observed when in combination with saturated fats. Here we discuss the current evidence relating dietary polyunsaturated fatty acids and enteric disease. 

## 11. Omega-6 Polyunsaturated Fatty Acids

As discussed above, the consumption of omega-6 polyunsaturated fatty acid is associated with an increase in pro-inflammatory microbes and host-inflammatory mediators. Not surprisingly, diets rich in omega-6 polyunsaturated fatty acids show robust inflammatory responses to enteric pathogens. While inflammation is required to survive infections, too much inflammation results in extensive host damage. Similar to diets rich in saturated fatty acids, post-natal diets rich in omega-6 polyunsaturated fatty acids results in exacerbated inflammatory-induced colonic damage during infection with *C. rodentium* [11,98]. At 10 days post-infection, mice fed diets rich in omega-6 polyunsaturated fatty acids had increased F480-positive macrophages and myeloperoxidase-positive neutrophils and higher inflammatory cytokines at the site of infection compared to mice fed fish oil. However, in contrast to diets rich in saturated fatty acids, omega-6 polyunsaturated fatty acids do not have microbial-driven protective responses [11]. In fact, the corn oil diet negatively correlated with the production of beneficial short-chain fatty acids which could be due to the observed increase in intestinal pathobionts [42]. Dietary omega-6 polyunsaturated fatty acids similarly show negative results in C57BL/6J mice injected intraperitoneally with LPS. As expected, mice fed omega-6 polyunsaturated fatty acid diets rich in sesame oil and soybean oil diets had marked production of the lipid mediators, prostaglandin E2 and leukotriene B4. However, this did not improve outcomes of infection and mice suffered an 80% mortality rate 24 and 48 h post-injection, respectively [97]. High omega-6 polyunsaturated fatty acids also worsen outcomes of diarrhea induced by rotavirus. Compared with diets rich in butter or olive oil, BALB/c pups fed corn oil rich in omega-6 polyunsaturated fatty acids had increased frequency of diarrhea and viral load when challenged with EDIM rotavirus [87]. In contrast, the negative effects of omega-6 polyunsaturated fatty acid-rich diets are not as evident in systemic *Salmonella* Typhimurium infections. While coconut oil diets rich in saturated fat were protective (88% survival) in Swiss Webster mice orally challenged with *Salmonella*, corn oil diets rendered a 62.5% survival rate. Arguably, this survival rate is low but it is substantially higher than the 48% survival rate observed in mice consuming fish oil rich in omega-3 polyunsaturated fatty acids [93]. Collectively, these data show that omega-6 polyunsaturated fatty acids have robust inflammatory responses whereby immune cells are effectively able to migrate to the site of inflammation and secrete pro-inflammatory mediators to eliminate the pathogen. However, this inflammatory cascade can result in excessive damage to the host resulting in an overall negative outcome. 

## 12. Omega-3 Polyunsaturated Fatty Acids

Long-chain omega-3 polyunsaturated fatty acids are hypothesized to improve several aspects of infant development [99]. As a result, most infant formula manufacturers now add manufactured eicosapentaenoic acid and docosahexaenoic acid to their product. However, several meta-analyses conclude that omega-3 polyunsaturated fatty acid supplements do not actually have an effect on infant visual acuity, memory, physical development, motor skills and cognition [100,101,102]. There is, however, a small amount of clinical evidence that omega-3 polyunsaturated fatty acid supplements may impair pro-inflammatory responses in infants [102]. Considering that inflammation is required to fight off infection, suppressing normal inflammatory responses in infants may have consequences during enteric infection. To this end, the addition of omega-3 polyunsaturated fatty acid supplements from fish oil to diets rich in corn oil increased populations of anti-inflammatory microbes such as *Lactobacillus* spp. and *Bifidobacterium* spp. in C57BL/6 mice, and attenuated pro-inflammatory responses [42]. While this resulted in less intestinal damage, the impaired inflammatory responses caused sepsis and significant mortality. Similar negative results were observed in *C. rodentium* infected C57BL/6 mice consuming omega-3 polyunsaturated fatty acids from flaxseed [103]. Omega-3 polyunsaturated fatty acids similarly exacerbated peritonitis-induced septic shock in rats [104]. 

The effects of omega-3 polyunsaturated fatty acids have also been investigated with respect to *Listeria monocytogenes* infections. Ingestion of Listeria by pregnant women can often result in fetal infection via transplacental transmission. Half of the newborns infected with *Listeria* die from the infection and therefore novel strategies to reduce the risk of mortality are being explored. Animal studies investigating the effects of dietary fatty acids on *Listeria* infections are conflicted and should be interpreted cautiously with attention paid to the experimental setup. Currently, there are two studies that report no effect of omega-3 polyunsaturated fatty acids on host resistance to *Listeria* infection [105,106]. One of these studies used (NZBxW)F_1_ mice defective in interleukin-12 biosynthesis [105,107], which may limit the predictive validity of this study. In contrast, a study using C3H/HeN mice found that four weeks of feeding with menhaden oil rich in omega-3 polyunsaturated fatty acids, delayed clearance and increased mortality from *Listeria* injected intraperitoneally [108]. These mice had lower serum levels of interleukin-12 and interferon-γ cytokines, which are important for the clearance of *Listeria.* This could explain the discrepancy with the previously mentioned (NZBxW)F_1_ study. Similarly, fish oil rich diets significantly reduced survival rates in BALB/c mice infected with *Listeria* through their tail vain [109] and resulted in a 67% mortality rate in C3H/HeN mice intraperitoneally injected with *Listeria*. For comparison, lard diets rich in monounsaturated fatty acids had 100% survival and soybean diets rich in omega-6 polyunsaturated fatty acids had a 58% survival rate [108]. 

The impacts of omega-3 polyunsaturated fatty acids on inflammatory or immune responses may also vary depending on the type of infection. Omega-3 polyunsaturated fatty acids had no impact on local intestinal inflammatory responses to infection with the parasite *Trichinella spiralis* in rats compared to an olive oil diet rich in monounsaturated fatty acids [110]. Overall, there are more papers published reporting adverse or no effects of omega-3 polyunsaturated fatty acids on host resistance to enteric challenge than those showing beneficial effects. However, the magnitude of these effects may be dependent on the background diet being consumed alongside the omega-3 polyunsaturated fatty acid supplements. For example, the addition of fish oil to milk fat diets induce protective responses during *C. rodentium* infection [11]. We theorize that this is why fish oil supplementation in infants consuming cows milk have altered gut microbial diversity but not infants consuming formula [111]. 

While this review focuses on enteric disease risk, the impacts of dietary fatty acids on other health areas should be considered. For instance, while omega-3 polyunsaturated fatty acids appear to have adverse or negligible impacts on enteric infections, they have been shown to be protective in animal studies on necrotizing enterocolitis [112,113] and *Pseudomonas aerginosa* lung infection [114], and there is clinical evidence that fish intake during pregnancy associates with lower allergic outcomes during infancy [115] but the results are conflicting [116]. Similarly, specialized pro-resolving mediators have been shown to improve outcomes of infection, but their role in the gut remains poorly understood. For instance, specialized pro-resolving mediators have been shown to improve outcomes of peritoneal *E. coli* infection [117] and severe influenza in mice [118]. In polymicrobial sepsis initiated by cecal ligation and puncture, Resolvin D2 increased survival in wild-type mice when compared with resolvin D2 knockout mice [119]. This indicates that specialized pro-resolving mediators should be considered in future investigations of enteric infections. Collectively, excessive intake of omega-6 polyunsaturated fatty acids appears to have negative health outcomes to enteric disease whereas the impact of omega-3 polyunsaturated fatty acids on health may be dependent on the other fats present. As omega-3 polyunsaturated fatty acids naturally make up a low percentage of breast milk fat, maintaining an appropriately low concentration of omega-3 polyunsaturated fatty acids in infant formula and including more saturated fatty acids similar to what is quantified in breast milk is recommended. 

## 13. Future Directions for Infant Formula

One of the greatest challenges for healthcare professionals involved in the prevention and treatment of diarrheal diseases is finding readily available, safe and cost-effective drug alternatives. While breast milk remains the best nutritional choice for infants, there are instances where breastfeeding is not possible, inadequate or unsuitable. The World Health Organization reports that only 40 percent of infants 6-months and under are breastfed exclusively [120]. In these instances, infant formulas are used as a breast-milk substitute. The formula market has grown into a global, multibillion-dollar industry [121,122,123]. While a great deal of resources has been devoted to improving the essential nutrients and energy requirements of formula, formula-fed infants continue to have increased risk of infection when compared with those exclusively breastfed. For instance, a recent study following 15,809 term infants in the United Kingdom showed that exclusive breastfeeding associated with reduced risk of chest infections and diarrhea in infants [124]. The failure of formula feeding to produce the same health benefits as breastfeeding could be due, at least in part, to fatty acid differences between the two nutritional sources. The Codex Alimentarius Commission, part of both the World Health Organization and the Agriculture Organization of the United Nations, governs the current global framework for infant formula. The Codex recommends a minimum and maximum amount of linoleic and alpha-linolenic acid and suggests that if docosahexaenoic acid is added to formula, arachidonic acid contents should reach the same concentration [125]. There are currently no recommendations for any monounsaturated or saturated fatty acids in the document, leaving the decision to include these fats up the manufacturers. Moreover, the Codex is inconsistently applied within countries. In Europe, regulation on the specific composition of infant formula has recently changed and the addition of docosahexaenoic acid but not arachidonic acid is now mandatory in infant formulas for full-term healthy neonates [126].

Legislation aside, there are several ways formula companies can reduce future disease risk in infants. A recent study in Singapore reported that formula-fed infants had consistently higher protein and lower total fat consumption compared to those who were breastfed [127]. This is in line with findings on Danish infants who similarly had lower fat intake in formula-fed infants compared to those who were breastfed [128]. Thus, increasing the total fat content in formula to mimic the composition in human milk should be a priority. In addition, formulas should be limited in the high percentages of omega-6 polyunsaturated fatty acids. In support of this, a study from Australia found that the dietary energy from polyunsaturated fats was higher for infants not breastfed [129]. Indeed, this year in France there has been a call to reduce the omega-6 polyunsaturated fatty acid, linoleic acid by blending formulas with dairy fat which have naturally low linoleic levels [73]. While most major formula companies still use non-fat milk and vegetable oil as a source of fat in their formulas, the addition of milk fat is gaining support [130,131]. For instance, the company Danone has added anhydrous milk fats to their Aptamil Profutura First Infant Milk formula [130] and several formulas in the European Union contain milk fat globule membrane, which is added to resemble the lipid profile of human milk [132]. Reintroducing animal milk fat in infant formula has other biological benefits. Since most commercially available formulas are based on vegetable oils, the sterol profile between human milk and formulas differ. By adding milk fat to formulas, companies can better mimic human milk composition and increase the bioaccessibility of cholesterol [133], which is an indispensable component of cell membranes. Milk fat is also a source of other bioactive lipids such as gangliosides and sphingomyelin, which are important in neuronal and brain development [132,134], and gut health [135]. Moreover, milk fat is a source of palmitic acid in the SN-2 position, which facilitates absorption [68,136]. These studies collectively show that animal sources of saturated fatty acids fulfill numerous metabolic and physiological functions. Therefore, formula companies are encouraged to consider the benefits of adding saturated milk fat to their products and limit the percentage of omega-6 polyunsaturated fatty acids, taking their lead from breast milk composition. 

## 14. Conclusions

While infant formula is intended to mimic the nutritional profile of human milk, there have been differences in the fatty acid profile between breast milk and formula, although formula companies continue to make improvements based on the best available evidence. The functional outcomes of any differences in nutrition during infancy are still largely unknown. It is increasingly apparent that the microbiome plays a key role in the establishment of the immune system; however, the consequences of dietary fat-induced modifications to the microbial communities during infancy are still being investigated. Here, we present evidence that different types of fatty acids result in different outcomes during enteric infection. Overall, saturated fatty acids protect against viral and bacterial agents and, like breast milk, should make up the largest fatty acid fraction of formula. While the strong inflammatory responses may initially cause epithelial damage, compensatory protective responses follow. The second-largest fatty acid fraction of formula should be composed of monounsaturated fatty acids. To date, no studies have yet shown any negative enteric health outcomes associated with monounsaturated fatty acids in the context of infection. Finally, polyunsaturated fatty acids should comprise a smaller proportion of formula fats. Evidence from animal models show that omega-6 polyunsaturated fatty acids have large inflammatory responses due to increased immune migration and expression of inflammatory cytokines at the site of infection. While inflammation is required to eliminate the pathogen, excessive inflammation causes epithelial damage and unlike saturated fats, omega-6 polyunsaturated fatty acids do not appear to have microbiome-driven compensatory responses, despite likely increases in downstream leukotrienes. In contrast, omega-3 polyunsaturated fatty acids and their metabolites have anti-inflammatory actions in experimental animal and culture studies. Omega-3 polyunsaturated fatty acids down-regulate cell adhesion molecules and reduce chemotaxis of inflammatory mediators to the site of infection. Once there, they decrease inflammatory cytokines and cause little overall inflammation. While this decreases mucosal damage in vivo and has mediators to resolve the damage, the host has increased risk of mortality because the lack of inflammation. This is observed when in combination with omega-6 polyunsaturated fatty acids. In contrast, omega-3 polyunsaturated fatty acids show no negative health consequences when combined with saturated fatty acids. Modifications to infant formula policies should reflect these findings and promote the re-introduction of milk fat, rich in saturated fatty acids. The benefits of re-introducing milk fat extend beyond saturated fatty acids and also introduce important bioactive lipids such as sphingomyelin and gangliosides which are not found in standard formulas. Given the interdependency of dietary fatty acids, immunity and the intestinal microbiota, a central challenge over the next few years will be to develop new experimental strategies to study this co-dependent network. Investigations clarifying this tripartite relationship are currently underway. 

## Figures and Tables

**Figure 1 nutrients-11-02064-f001:**
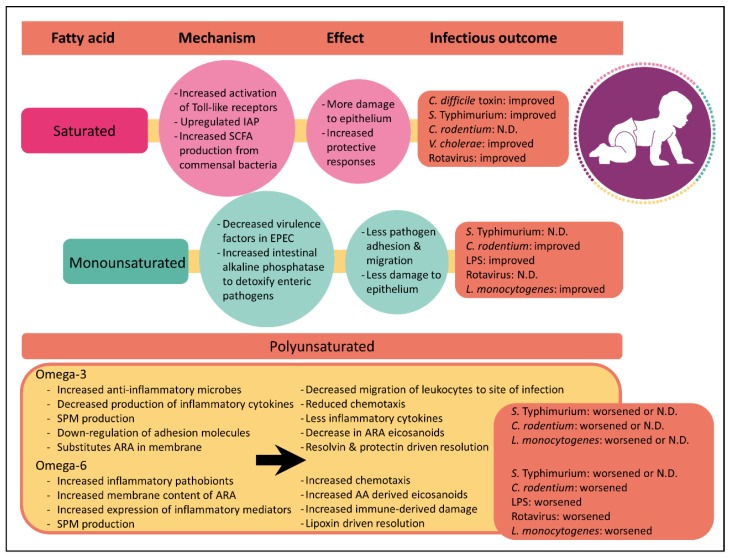
Summary of mechanistic insights, inflammatory effects and enteric disease outcomes driven by saturated, monounsaturated and polyunsaturated fatty acids. Improved, worsened or no difference (N.D) describes the severity of diarrheal infectious outcomes when compared to other diets tested. If two studies had conflicting results, both outcomes are listed. Abbreviations: IAP, intestinal alkaline phosphatase; SPM, specialized pro-resolving lipid mediators; SCFA, short-chain fatty acids; LPS, lipopolysaccharide; ARA, arachidonic acid; EPEC, Enteropathogenic *Escherichia coli*.

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
