# Peer review of "Dietary Fatty Acids and Host–Microbial Crosstalk in Neonatal Enteric Infection"

_nutrients, 2019, doi:10.3390/nu11092064_

Round 1
Reviewer 1 Report
This is an interesting manuscript with a creative approach for the infant nutrition field. The paper aims to describe the role of dietary fatty acids in modulating host inflammation, both directly and through the microbiome. Nevertheless, the manuscript suffers from a few limitations. In what follows, I will elaborate on them trying to provide solutions.
My main concerns with this paper are:
Most of the manuscript is about fatty acids and not dietary lipids. I like the idea behind this paper and I agree with the conclusion that altering the lipid composition in infant formula may present a potential strategy to improve infections outcomes in formula fed infants. However, this review is focused on fatty acids, it is mentioning briefly the cholesterol and there is little described about other dietary lipids such as complex lipids (i.e., phospholipids and gangliosides). I would strongly suggest authors to substitute dietary lipids by fatty acids all over the manuscript including the title; otherwise, the positioning of the paper would not be clear at all. The section about future directions for Infant formula is overstressing a problem with infant formulas without providing solid arguments. Some examples follow: Pp 11, Line 483-484: “great deal of resources…short term safety….the issue of its nutritional adequacy has not been reached” I disagree about that sentence showing infant formulas are not nutritionally adequate and there has not been any progress on research in the last decades. Why nutritional adequacy is defined in this paper as “whereas nutritional adequacy considers long term outcomes such as lifelong immune functions”? What is the scientific definition of nutritional adequacy? Please clarify these issues. Pp 11, Lines 498-510: I suggest to take out the Nestle example. Your paper is a review and I believe it doesn’t make sense to add such an example and based on it, offer recommendations for future directions on infant formula. Nestle is not the only brand in US and it is not representing the most important formula companies around the world. You need to be careful with these issues. Pp 11, Line 517: “The EU has not yet approved re-introducing cow’s milk fat in infant formula”. There is no restriction to use dairy fat in infant formulas in Europe. Please check again EU legislation. Table 1. I suggest to remove table 1. The saturated fat data from mature milk in this table does not represent the mean values of saturated fat from other studies. e.g: Guerra, E., Downey, E., O'Mahony, J. A., Caboni, M. F., O'Shea, C. A., Ryan, A. C., & Kelly, A. L. (2016). Influence of duration of gestation on fatty acid profiles of human milk. European journal of lipid science and technology, 118(11), 1775-1787. Gibson, R. A., & Kneebone, G. M. (1981). Fatty acid composition of human colostrum and mature breast milk. The American journal of clinical nutrition, 34(2), 252-257. Lopez-Lopez, A., Lopez-Sabater, M. C., Campoy-Folgoso, C., Rivero-Urgell, M., & Castellote-Bargallo, A. I. (2002). Fatty acid and sn-2 fatty acid composition in human milk from Granada (Spain) and in infant formulas. European journal of clinical nutrition, 56(12), 1242. Pp12, Lines 542-544: Your proposal is solely based on Nestle’s policy on saturated fats. As argued earlier, Nestle is not the only infant food company and therefore you have to be extremely careful when making this statement for the whole industry. I suggest to eliminate it.Other issues
Introduction:
Pp 2, Line 46: “formula companies derive their fatty acids from vegetable sources which differ greatly from the fatty acids found in human breast milk (Delplanque et al., 2015). I suggest updating this info and check what is happening today. Infant formula producers usually use a mixture of vegetable oils to mimic human milk fatty acid profile and in the last 5 years dairy fat has been widely used. Please see:
Hageman, J. H., Danielsen, M., Nieuwenhuizen, A. G., Feitsma, A. L., & Dalsgaard, T. K. (2019). Comparison of bovine milk fat and vegetable fat for infant formula: Implications for infant health. International dairy journal, 92, 37-49. Claumarchirant, L., Cilla, A., Matencio, E., Sanchez-Siles, L. M., Castro-Gomez, P., Fontecha, J., ... & Lagarda, M. J. (2016). Addition of milk fat globule membrane as an ingredient of infant formulas for resembling the polar lipids of human milk. International Dairy Journal, 61, 228-238.Future direction for Infant formula
Pp 12, Lines 529-540: I would try to simplify the benefits of using animal milk fat in the design of infant formulas. As an example: 1) As a source of palmitic acid in SN-2 position which facilitates absorption 2) As a source of cholesterol (e.g. Hamdam et al., 2018). I would also add being a source of other bioactive lipids such as Gangliosides (e.g. Plamano et al., 2015) and Sphingomyelin (e.g. Claumarchirant et al., 2016)
Hamdan, I. J., Sanchez-Siles, L. M., Garcia-Llatas, G., & Lagarda, M. J. (2018). Sterols in infant formulas: a bioaccessibility study. Journal of agricultural and food chemistry, 66(6), 1377-1385. Palmano, K., Rowan, A., Guillermo, R., Guan, J., & McJarrow, P. (2015). The role of gangliosides in neurodevelopment. Nutrients, 7(5), 3891-3913.Conclusion:
Pp 13, Lines 575-577: Suggest to rephrase the benefits of using Cow’s milk fat. Please take in mind that the content of omega 6 is not a key factor to use cow’s milk in the design of infant formulas. Final omega 6 content in infant formulas can be modified by using different types and quantities of vegetable oils and the use of cow’s milk is not determinant to achieve adequate omega 6 levels.
Good luck with your research!
Author Response
Reviewer 1:
This is an interesting manuscript with a creative approach for the infant nutrition field. The paper aims to describe the role of dietary fatty acids in modulating host inflammation, both directly and through the microbiome. Nevertheless, the manuscript suffers from a few limitations. In what follows, I will elaborate on them trying to provide solutions.
My main concerns with this paper are:
Comment 1: Most of the manuscript is about fatty acids and not dietary lipids. I like the idea behind this paper and I agree with the conclusion that altering the lipid composition in infant formula may present a potential strategy to improve infections outcomes in formula fed infants. However, this review is focused on fatty acids, it is mentioning briefly the cholesterol and there is little described about other dietary lipids such as complex lipids (i.e., phospholipids and gangliosides). I would strongly suggest authors to substitute dietary lipids by fatty acids all over the manuscript including the title; otherwise, the positioning of the paper would not be clear at all.
Response 1: We thank the reviewer for their comments, which have helped us to improve the quality and clarity of this review. As suggested, we substituted dietary lipids with fatty acids all over the manuscript to clarify the positioning of the paper.
Comment 2: The section about future directions for Infant formula is overstressing a problem with infant formulas without providing solid arguments. Some examples follow: Pp 11, Line 483-484: “great deal of resources…short term safety….the issue of its nutritional adequacy has not been reached” I disagree about that sentence showing infant formulas are not nutritionally adequate and there has not been any progress on research in the last decades. Why nutritional adequacy is defined in this paper as “whereas nutritional adequacy considers long term outcomes such as lifelong immune functions”? What is the scientific definition of nutritional adequacy?
Response 2: Nutritional adequacy is defined as the sufficient intake of essential nutrients, needed to fulfill nutritional requirements for optimal health {Castro-Quezada, 2014 #498}. Since breastfed infants consistently have lower incidences of enteric disease, we argue that formula does not provide optimal health (Kramer and Kakuma, 2004; Quigley et al., 2007). However, we agree that it may be overstressing a problem and as such, we have toned down the wording and provided more solid arguments. This section now reads, “While a great deal of resources has been devoted to improving the essential nutrients and energy requirements of formula, formula-fed infants continue to have increased risk of infection when compared with those exclusively breastfed. For instance, a recent study following 15, 809 term infants in the United Kingdom showed that exclusive breastfeeding associated with reduced risk of chest infections and diarrhoea in infants (Quigley et al., 2016).” Line 571-575
Comment 3: Please clarify these issues. Pp 11, Lines 498-510: I suggest to take out the Nestle example. Your paper is a review and I believe it doesn’t make sense to add such an example and based on it, offer recommendations for future directions on infant formula. Nestle is not the only brand in US and it is not representing the most important formula companies around the world. You need to be careful with these issues.
Response 3: We thank the reviewer for this constructive feedback and have re-wrote this section to include a broader overview of formulas and have removed the table.
Comment 4: Pp 11, Line 517: “The EU has not yet approved re-introducing cow’s milk fat in infant formula”. There is no restriction to use dairy fat in infant formulas in Europe. Please check again EU legislation.
Response 4: As requested, this section has been rephrased.
Comment 5: Table 1. I suggest to remove table 1. The saturated fat data from mature milk in this table does not represent the mean values of saturated fat from other studies. e.g: Guerra, E., Downey, E., O'Mahony, J. A., Caboni, M. F., O'Shea, C. A., Ryan, A. C., & Kelly, A. L. (2016). Influence of duration of gestation on fatty acid profiles of human milk. European journal of lipid science and technology, 118(11), 1775-1787. Gibson, R. A., & Kneebone, G. M. (1981). Fatty acid composition of human colostrum and mature breast milk. The American journal of clinical nutrition, 34(2), 252-257. Lopez-Lopez, A., Lopez-Sabater, M. C., Campoy-Folgoso, C., Rivero-Urgell, M., & Castellote-Bargallo, A. I. (2002). Fatty acid and sn-2 fatty acid composition in human milk from Granada (Spain) and in infant formulas. European journal of clinical nutrition, 56(12), 1242. Pp12, Lines 542-544: Your proposal is solely based on Nestle’s policy on saturated fats. As argued earlier, Nestle is not the only infant food company and therefore you have to be extremely careful when making this statement for the whole industry. I suggest to eliminate it.
Response 5: As requested, we have eliminated Table 1 and re-worded the future directions for infant formula section.
Other issues
Introduction:
Comment 6: Pp 2, Line 46: “formula companies derive their fatty acids from vegetable sources which differ greatly from the fatty acids found in human breast milk (Delplanque et al., 2015). I suggest updating this info and check what is happening today. Infant formula producers usually use a mixture of vegetable oils to mimic human milk fatty acid profile and in the last 5 years dairy fat has been widely used. Please see:
Hageman, J. H., Danielsen, M., Nieuwenhuizen, A. G., Feitsma, A. L., & Dalsgaard, T. K. (2019). Comparison of bovine milk fat and vegetable fat for infant formula: Implications for infant health.
International dairy journal, 92, 37-49. Claumarchirant, L., Cilla, A., Matencio, E., Sanchez-Siles, L. M., Castro-Gomez, P., Fontecha, J., ... & Lagarda, M. J. (2016). Addition of milk fat globule membrane as an ingredient of infant formulas for resembling the polar lipids of human milk. International Dairy Journal, 61, 228-238.
Response 6: We agree that formula companies are slowly moving towards adding milk fat to their products; however, the majority of them still use non-fat milk and vegetable oils. Nestle is the global market leader followed by the Swiss company, Danone and Rickitt Benckiser Group Plc, which acquired Mead Johnson in 2017. Collectively, these manufacturers account for 56% of the formula market (Kent, 2015). While Danone has a product on the market using milk fat (Aptamil Profutura first infant milk), to the best of our knowledge Nestle does not. Enfamil newborn similarly uses non-fat milk and vegetable oil as a source of fat. However, we acknowledge that there are other formulas on the market and we agree that studies have shown milk fat globule membrane in formulas from Europe, Asia and America. We have updated our manuscript to reflect this in the Future directions for infant formula section (lines 598-605).
Future direction for Infant formula
Comment 7: Pp 12, Lines 529-540: I would try to simplify the benefits of using animal milk fat in the design of infant formulas. As an example: 1) As a source of palmitic acid in SN-2 position which facilitates absorption 2) As a source of cholesterol (e.g. Hamdam et al., 2018). I would also add being a source of other bioactive lipids such as Gangliosides (e.g. Plamano et al., 2015) and Sphingomyelin (e.g. Claumarchirant et al., 2016)
Hamdan, I. J., Sanchez-Siles, L. M., Garcia-Llatas, G., & Lagarda, M. J. (2018). Sterols in infant formulas: a bioaccessibility study. Journal of agricultural and food chemistry, 66(6), 1377-1385. Palmano, K., Rowan, A., Guillermo, R., Guan, J., & McJarrow, P. (2015). The role of gangliosides in neurodevelopment. Nutrients, 7(5), 3891-3913.
Response 7: We thank the reviewer for this constructive feedback and have changed this section to reflect their suggestions. The section now reads, “Reintroducing animal milk fat in infant formula has other biological benefits. Since most commercially available formulas are based on vegetable oils, the sterol profile between human milk and formulas differ. By adding milk fat to formulas, companies can better mimic human milk composition and increase the bioaccessibility of cholesterol (Hamdan et al., 2018), which is an indispensable component of cell membranes. Milk fat is also a source of other bioactive lipids such as Gangliosides and Sphingomyelin, which are important in neuronal and brain development (Palmano et al., 2015; Claumarchirant et al., 2016), and gut health (Nilsson, 2016). Moreover, animal milk fat is a source of palmitic acid in the SN-2 position, which facilitates absorption (Bar-Yoseph et al., 2016; Mendonca et al., 2017). These studies collectively show that animal sources of saturated fatty acids fulfill numerous metabolic and physiological functions. Therefore, formula companies are encouraged to consider the benefits of adding milk fat to their products and limit the percentage of omega-6 polyunsaturated fatty acids, taking their lead from breast milk composition.”
Conclusion:
Comment 8: Pp 13, Lines 575-577: Suggest to rephrase the benefits of using Cow’s milk fat. Please take in mind that the content of omega 6 is not a key factor to use cow’s milk in the design of infant formulas. Final omega 6 content in infant formulas can be modified by using different types and quantities of vegetable oils and the use of cow’s milk is not determinant to achieve adequate omega 6 levels.
Good luck with your research!
Response 8: Thank you! We have re-phrased this section for clarity.
Reviewer 2 Report
The review article by Quin and Gibson focuses on data to show that enteric infections are potentially influenced by the composition of dietary fatty acids and their crosstalk with the gut microbiome. The implications of the work are on development of infant formulas and future recommendations on the composition of dietary fat in the formulas. However, there are some concerns that need to be addressed:
1. Please add a statement between lines 127-158 that many of the rodent microbiome studies focused on the effects of different fatty acids rely on concentrations of SFAs, MUFAs, and PUFAs that are not physiological. Alternatively, the authors could just add a separate section stating that a major limitation of the rodent studies is that investigators often rely on doses of differing oils that do not represent human intake. This limitation is critical to point out. Examples can also be given if needed.
2. Line 159- fix minor grammar issue.
3. Lines 171 and below. Fix left justification.
4. Lines 171-203. The section on fatty acids and inflammation is outdated and is not in agreement with the current view. The authors need to correct the following in this section and add the appropriate references listed below:
a) The notion that n-3 PUFAs are “anti-inflammatory” is oversimplified and needs to be updated. The authors need to acknowledge that LC n-3 PUFAs EPA and DHA are not just anti-inflammatory but exert pro-resolving effects by serving as precursors for the production of specialized pro-resolving lipid mediators (SPMs) (PMID: 29757195; other Serhan references can also be added if needed).
b) Next, please acknowledge that n-3 PUFAs and their downstream metabolites SPMs can have strong effects on the outcomes to differing infections (PMID: 23477864; 28500069; 27994074), yet the role of SPMs on enteric infections remains unexplored. In fact, the authors can come back to this point when they elegantly point out negative effects of fish oils on several enteric infections later in the review and can point to the fact that there is a need to study SPMs, rather than fish oils, on the host microbiome and gut infections.
c) The notion that n-6 PUFAs are pro-inflammatory needs to be clarified. Please note that derivatives of lipoxin A4 (synthesized from arachidonic acid) are pro-resolving and also have a role in infection (PMID: 29523657), although their role in the gut remains poorly explored.
5. Figure 1 should be updated to reflect that select omega-3 and omega-6 fatty acids are precursors for SPMs that regulate inflammation resolution.
6. Line 322. It should be pointed out that hydrogenated fat has its own risks associated with a variety of diseases including CVD.
7. The section on fish oil (lines 377 and below) should point out that rodent studies that rely on fish oils have major limitations, notably there are other components in the oils that may also influence the microbiome. Finally, acknowledge that while dietary omega-3s may be important for brain development and inflammation (see work of RP Bazinet), there may be negative effects that extend beyond infection and potentially gut cancers (PMID: 20798218).
8. Line 568 – again, please correct the notion on omega-3s as anti-inflammatory to omega-3s and their metabolites have anti-inflammatory and pro-resolving properties.
Author Response
Plase see the attachment

Round 2
Reviewer 1 Report
I appreciate the great efforts that the authors have made in response to
my questions and concerns. I think the manuscript has been significantly improved.